# Hemagglutinin Subtype Specificity and Mechanisms of Highly Pathogenic Avian Influenza Virus Genesis

**DOI:** 10.3390/v14071566

**Published:** 2022-07-19

**Authors:** Anja C. M. de Bruin, Mathis Funk, Monique I. Spronken, Alexander P. Gultyaev, Ron A. M. Fouchier, Mathilde Richard

**Affiliations:** 1Department of Viroscience, Erasmus Medical Center, 3000 CA Rotterdam, The Netherlands; a.c.m.debruin@erasmusmc.nl (A.C.M.d.B.); m.funk@erasmusmc.nl (M.F.); m.spronken@erasmusmc.nl (M.I.S.); a.goultiaev@erasmusmc.nl (A.P.G.); r.fouchier@erasmusmc.nl (R.A.M.F.); 2Group Imaging and Bioinformatics, Leiden Institute of Advanced Computer Science (LIACS), Leiden University, 2300 RA Leiden, The Netherlands

**Keywords:** Highly Pathogenic Avian Influenza Viruses, Low-Pathogenic Avian Influenza Viruses, multibasic cleavage site, virulence, pathogen evolution, proteolytic cleavage, RNA-dependent RNA polymerase

## Abstract

Highly Pathogenic Avian Influenza Viruses (HPAIVs) arise from low pathogenic precursors following spillover from wild waterfowl into poultry populations. The main virulence determinant of HPAIVs is the presence of a multi-basic cleavage site (MBCS) in the hemagglutinin (HA) glycoprotein. The MBCS allows for HA cleavage and, consequently, activation by ubiquitous proteases, which results in systemic dissemination in terrestrial poultry. Since 1959, 51 independent MBCS acquisition events have been documented, virtually all in HA from the H5 and H7 subtypes. In the present article, data from natural LPAIV to HPAIV conversions and experimental in vitro and in vivo studies were reviewed in order to compile recent advances in understanding HA cleavage efficiency, protease usage, and MBCS acquisition mechanisms. Finally, recent hypotheses that might explain the unique predisposition of the H5 and H7 HA sequences to obtain an MBCS in nature are discussed.

## 1. Introduction

Avian influenza A viruses (AIVs) circulate in wild waterfowl, predominantly in the orders of the *Anseriformes* (e.g., ducks and geese) and *Charadriiformes* (e.g., gulls) [1]. AIVs are categorized based on the antigenic properties of their surface glycoproteins, the hemagglutinin (HA) and the neuraminidase (NA). Sixteen subtypes of HA (H1–H16) and nine subtypes of NA (N1–N9) have been distinguished in wild waterfowl [2]. In wild waterfowl, AIV infections are generally asymptomatic and do not cause histological lesions [3,4]. The replication of AIVs primarily takes place in the gastro intestinal tract (GIT), which results in fecal shedding with limited oropharyngeal shedding [5,6,7]. Therefore, fecal–oral transmission is considered to be the main route of transmission among wild birds.

Incursions of AIVs from wild waterfowl into terrestrial poultry species, e.g., chickens and turkeys, occur frequently via (in)direct contact [8,9]. In these species, AIV replication is restricted to the GIT and respiratory tract (RT), resulting in mild clinical manifestations. Such AIVs are called Low Pathogenic Avian Influenza Viruses (LPAIVs) [10]. Upon transmission to terrestrial poultry, LPAIVs of the H5 and H7 subtypes can mutate into Highly Pathogenic Avian Influenza Viruses (HPAIVs). HPAIVs cause severe hemorrhagic disease with mortality rates reaching 100% and therefore pose a significant threat to animal welfare and to the poultry industry [10]. The LPAIV to HPAIV transition is characterized by the acquisition of nucleotides coding for basic amino acids at the proteolytic cleavage site of HA. Such nucleotides can be acquired through nucleotide substitutions, sequence duplications, and/or non-homologous recombination (NHR) between HA RNA and RNA from viral or host origin. The presence of multiple basic amino acids at the cleavage site of HA, the multibasic cleavage site (MBCS), is the main determinant of the high virulence of HPAIVs in terrestrial poultry [10]. Official guidelines for the classification of AIVs as LPAIV or HPAIV have been established by The Organization of Animal Health (OIE) [11]. Classification as HPAIV is determined by the pathogenicity upon intravenous inoculation of chickens, and/or the amino acid sequence at the cleavage site of HA. Pathogenicity is defined by the Intravenous Pathogenicity Index (IVPI), which has to be experimentally determined by following disease progression in 10 4- to 8-week-old chickens at 24-h intervals for 10 days and scoring each bird as follows: 0 if normal, 1 if sick, 2 if severely sick, or 3 if dead. The IVPI is the mean score per bird per observation over the 10-day period. AIVs that have an IVPI between 1.2 and 3 are classified as HPAIV. AIVs that contain an MBCS with an identical amino acid sequence as that of previously identified HPAIVs are considered highly pathogenic irrespective of experimentally confirmed pathogenicity. The IVPI has to be determined for AIVs with novel MBCS motifs to allow for HPAIV classification.

There have been 51 independent documented events of MBCS acquisition in natural settings since 1959. In this review, all historical emergences of AIVs with an MBCS since 1959 are listed, and the current understanding of MBCS amino acid composition and acquisition is discussed. Furthermore, we shine light on the remaining questions regarding HPAIV genesis, primarily on the molecular mechanisms of nucleotide insertions leading to MBCS acquisition and why those are restricted to LPAIVs of the H5 and H7 subtypes.

## 2. Historical Emergences of HPAIVs

The first recorded AIV outbreak occurred in 1878, when a chicken flock in Italy contracted a contagious disease with high mortality rates [12]. Over the following decades, outbreaks with similar clinical manifestations occurred in poultry flocks and the causative pathogen was termed Fowl Plague Virus. These outbreaks were later determined to be caused by HPAIVs of the H7 subtype [13]. The first H5 HPAIV was isolated in 1959 in Scotland [14], after which the causative agents of the many HPAIV outbreaks that followed were determined and subtyped. All reported AIVs with MBCS since 1959 are listed in Table 1 (also reviewed in [15,16]). Their epidemiology varied as some caused outbreaks and others were single detections. Interestingly, not all MBCS-containing AIVs had a highly pathogenic phenotype, therefore Table 1 includes AIVs of high, low, and unknown pathogenicity. The only non-H5/H7 AIV with an MBCS is an H4N2 LPAIV from 2012 [17]. The number of MBCS acquisitions in AIVs per year has increased over time (Figure 1A), which correlates with improved surveillance efforts and the increasing demand for poultry in an expanding human population. The highest number of reported MBCS acquisitions in AIVs occurred in Europe and North America (Figure 1B). Phylogenetic analysis of newly emerged MBCS-containing AIVs showed that they belong to unique lineages [15,18], indicating that they arose from independent LPAIV to HPAIV conversions.

Generally, HPAIV outbreaks occurred in terrestrial poultry holdings, e.g., chicken, turkey, and ostrich farms, and were short-term epizootics with limited regional spread. Most outbreaks were halted by stringent culling measures and/or vaccination. However, despite intervention attempts, multiple HPAIVs have circulated for longer periods of time following their initial emergence. HPAIVs from the H5 lineage whose HA originates from the H5N1 HPAIV A/goose/Guangdong/1/96 (Gs/Gd) strain are currently enzootic in poultry and wild birds in various parts of the world. Gs/Gd-H5 HPAIVs caused outbreaks in poultry in Hong Kong in 1997, alongside human infections [19,20]. The outbreaks were stamped out by a massive poultry eradication scheme. However, the Gs/Gd-H5 HPAIVs resurfaced in terrestrial and aquatic poultry and wild birds in Hong Kong and mainland China in 2001/2002 and caused two human infections in Hong Kong in 2003 [21]. From late 2003 onwards, Gs/Gd-H5 HPAIVs spread to many countries on four continents, likely through poultry trade and bird migration, causing outbreaks in poultry and wild birds and sporadic human infections [22,23,24]. As of the 4th of March 2022, a total of 953 confirmed human cases of Gs/Gd-H5 HPAIV infection have been documented, of which 491 were fatal, emphasizing the threat that HPAIVs pose to human health [25]. In 2012, H7N3 HPAIVs arose in Mexico and became enzootic in poultry populations despite vaccination efforts [26]. Only two human cases of conjunctivitis due to the Mexican H7N3 HPAIVs have been reported in almost a decade of circulation [27]. From 2013 to 2018, H7N9 AIVs caused annual waves of human infection in China. Sporadic human-to-human transmission was reported, increasing the concern for a potential pandemic [28]. Initially, the H7N9 AIVs were LPAIVs. In 2016, H7N9 HPAIVs were detected following an LPAIV to HPAIV conversion in chickens [29,30,31]. Poultry vaccination programs were put in place, which greatly reduced H7N9 AIV detections but did not eliminate circulation entirely [32,33]. As of the 4th of March 2022, the total number of reported zoonotic H7N9 infections was 1568, including 616 deaths [25].

**Table 1 viruses-14-01566-t001:** Historical emergence of MBCS-containing AIVs 1959–2021: epidemiological data, cleavage site sequence, IVPI, phenotypical classification, and progenitor (LPAIV) detection.

Year	Country	Subtype	Cleavage Site Sequence	HA Accession Number	IVPI (Phenotype ^a^)	Progenitor Detection ^b^	HA Accession Number Progenitor	Number of Affected Premises (Species)	Ref.
**1959**	Scotland	H5N1	PQRKKR*G ^c^	GU052518	N.D. ^d^ (HP)	No ^e^	N.A. ^f^	2 (unknown)	[14]
**1961**	South Africa	H5N3	PQRETRRQKR*G	GU052822	N.D. (HP)	No	N.A.	N.A. ^g^ (common terns)	[34,35,36]
**1963**	England	H7N3	PKRRRR*G	AF202238	N.D. (HP)	No	N.A.	3 (turkeys)	[37,38]
**1966**	Canada	H5N9	PQRRKKR*G	CY107859	N.D. (HP)	Yes	CY087808	1 (turkeys)	[39,40]
**1967**	USSR	H5N1	unknown	N.A.	N.D. (HP)	No	N.A.	1 (chickens)	[41]
**1976**	Australia	H7N7	PEIPKKKEKR*G	CY024786	N.D. (HP)	No ^h^	N.A.	1 (chickens)	[42,43]
**1979**	England	H7N7	PEIPKKRKR*G PEIPKRRRR*G PEIPKKREKR*G	N.A.	N.D. (HP)	No	N.A.	3 (turkeys)	[44]
**1979**	Germany	H7N7	PEIPKKKKR*G PEIPKRKKR*G PEIPKKKKKKR*G PEIPKKRKKR*G PETPKKKKKKR*G	U20459 L43913 CY107844 L43915 L43914	N.D. (HP)	No ^i^	N.A.	2 (chickens, geese)	[45]
**1983–1984**	USA	H5N2	PQKKKR*G	GU052771	2.37 (HP)	Yes	J04325	356 (chickens, turkeys, guinea fowl, chuckar)	[46,47,48]
**1983**	Ireland	H5N8	PQRKRKKR*G	M18451	2.76 (HP)	No	N.A.	4 (turkeys, ducks, chickens)	[49]
**1985**	Australia	H7N7	PEIPKKREKR*G	M17735	N.D. (HP)	No	N.A.	1 (chickens)	[50,51]
**1991**	England	H5N1	PQRKRKTR*G	GU052510	3 (HP)	No	N.A.	1 (turkeys)	[52]
**1992**	Australia	H7N3	PEIPKKKKR*G	AF202227	2.71 (HP)	No	N.A.	1 (chickens, ducks)	[53,54]
**1994–1995**	Australia	H7N3	PEIPRKRKR*G	CY022685	N.D. (HP)	No	N.A.	1 (chickens)	[54]
**1994–1995**	Mexico	H5N2	PQRKRKTR*G PQRKRKRKTR*G	AB558473 U85390	N.D. (HP)	Yes	GU186573	360 (chickens)	[55,56,57]
**1994–1995**	Pakistan	H7N3	PETPKRKRKR*G PETPKRRKR*G	AF202226 AF202230	N.D. (HP)	No	N.A.	Many (chickens)	[58,59]
**1996**– **now**	China	H5N1 + H5Nx ^j^	PQRERRRKKR*G ( + many variations)	AF144305	2.1 (HP)	No	N.A.	Many (many)	[60]
**1997**	Australia	H7N4	PEIPRKRKR*G PEIPRKRKR*G	AY943924 CY022701	2.52 ^k^ (HP)	No	N.A.	3 (chickens, emu)	[61]
**1997–1998**	Italy	H5N2	PQRRRKKR*G	CY017403	2.98–3 (HP)	No	N.A.	8 outbreaks (chickens, ducks, geese, turkeys, guinea fowl, pigeons)	[62]
**1999–2000**	Italy	H7N1	PKGSRVRR*G	CY021405	3 (HP)	Yes	GU052999	413 outbreaks (chickens, turkeys, guinea fowl, ducks, pheasants, quails, ostriches)	[63,64]
**2002**	Chile	H7N3	PEKPKTCSPLSRCRETR*G PEKPKTCSPLSRCRKTR*G	AY303631 AY303632	2.24–3 (HP)	Yes	AY303630	4 (chickens, turkeys)	[65,66,67]
**2003**	Netherlands	H7N7	PEIPKRRRR*G	AY338458	2.93 (HP)	No ^l^	N.A.	255 (chickens)	[68,69]
**2003**	Netherlands	H7N3	PEIPKGSRVRR*G	EPI1595425	2.4 (HP)	Yes	EPI1595417	N.A. ^m^ (turkeys)	[70,71]
**2003**	Pakistan	H7N3	PETPKRRKR*G	HM346493	2.8 (HP)	No	N.A.	522 (chickens)	[59,72]
**2004**	USA	H5N2	PQRKKR*G	AY849793	0 (LP)	No	N.A.	6 (chickens)	[73]
**2004**	Canada	H7N3	PENPKQAYRKRMTR*G (+ many variations)	AY648287	2.87 (HP)	Yes	AY650270	42 (chickens)	[74]
**2004**	South Africa	H5N2	PQREKRRKKR*G	FJ519983	0.63 ^n^ (HP)	No	N.A.	38 (ostriches)	[75]
**2005**	North Korea	H7N7	PEIPKGRHRRPKR*G	N.A.	N.D. (HP)	No	N.A.	3 (chickens	[76]
**2006**	South Africa	H5N2	PQRRKKR*G	EF591749	N.D. (HP)	Yes	EF591757	24 (ostriches)	[77]
**2007**	Canada	H7N3	PENPKTTKPRPRR*G	EU500860	3 (HP)	No	N.A.	1 (chickens)	[78]
**2007**	Nigeria	H5N2	KEKRRKKR*G REKRRKKR*G	N.A.	N.D. (LP^o^)	No	N.A.	N.A. ^o^ (duck, geese)	[79]
**2008**	England	H7N7	PEIPKRKKR*G PEIPKKKKR*G PEIPKKKKKKR*G	FJ476173	2.95–3 (HP)	No	N.A.	1 (chickens)	[80]
**2009**	Spain	H7N7	PKGTKPRPRR*G	GU121458	N.D. (HP)	No	N.A.	1 (chickens)	[81]
**2011–2013**	South Africa	H5N2	PQRRKKR*G PQRRRKKR*G PQRKRKKR*G PQRRRKR*G	JX069081	0.8–1.37 ^p^ (HP)	No	N.A.	50 (ostriches)	[82]
**2012**	Australia	H7N7	PEIPRKRKR*G	N.A.	N.D. (HP)	No	N.A.	1 (chickens)	[83]
**2012–2013**	Taiwan	H5N2	PQRKKR*G PQRRKR*G	KJ162620 KF193394	2.91 (HP)	No ^q^	N.A.	5 (chickens)	[84]
**2012–now**	Mexico	H7N3	PENPKDRKSRHRRTR*G	JX908509	N.D. (HP)	No	N.A.	Many (chickens)	[85,86]
**2012**	USA	H4N2	PEKRRTR*G	KF986862	0 (LP)	No	N.A.	1 (quail)	[17]
**2013**	Italy	H7N7	PKRKRR*G PKRKRR*G PKRRERR*G PKRKRR*G	KF569186	N.D. (HP)	No	N.A.	6 (chickens, turkeys)	[87,88]
**2013**	Australia	H7N2	PEIPRKRKR*G	N.A.	N.D. (HP)	No	N.A.	2 (chickens)	[89]
**2015**	England	H7N7	PEIPRHRKGR*G PEIPRHRKRR*G	EPI623939	2.52 (HP)	No	N.A.	1 (chickens)	[90]
**2015**	Germany	H7N7	PEIPKRKRR*G	EPI634885	N.D. (HP)	Yes	EPI624526	1 (chickens)	[91]
**2015–2016**	France	H5(N1, N2,N9)	HQRRKR*G	H5N1:KU310447 H5N2:KX014878 H5N9:KX014886	H5N1: 2.9 (HP)	No	N.A.	81 (chickens, ducks, geese, guinea fowl)	[92,93]
**2016**	Algeria	H7N1	unknown	N.A.	N.D. (HP)	No	N.A.	N.A. ^r^ (many wild bird species)	[94]
**2016**	USA	H7N8	PKKRKTR*G	KU558906	N.D. (HP)	Yes	EPI709576	1 (turkeys)	[95,96]
**2016**	Italy	H7N7	PELPKGRKRR*G PELPKRRERR*G	EPI756028	N.D. (HP)	No	N.A.	2 (chickens, turkeys)	[97]
**2016–now**	China	H7N9	PEVPKGKRTAR*G PEVPKRKRTAR*G PEVPKGKRIAR*G	EPI919533 EPI917102	2.92 (HP)	No ^s^	N.A.	Many (primarily chickens)	[29,30]
**2017**	USA	H7N9	PENPKTDRKSRHRRIR*G	MF357740	N.D. (HP)	Yes	MF357732	2 (chickens)	[98]
**2020**	USA	H7N3	PENPKTDRKSRHRRIR*G	EPI1775733	2.46 (HP)	Yes	MT444363	13 (turkeys)	[99]
**2020**	Australia	H7N7	unknown	N.A.	N.D. (HP)	No	N.A.	3 (chickens)	[100]
**2021**	Lithuania	H7N7	unknown	N.A.	N.D. (HP)	No	N.A.	N.A.^t^ (mute swan)	[101]

^a^ Phenotype of viral strain is based on whether the birds were showing severe (HP) or no (LP) symptoms during the time of detection and/or IVPI. ^b^ Direct progenitor is defined as the LPAIV that was detected in the same or neighboring poultry farm prior to HPAIV detection. If a virus with a closely related HA gene was reported in either poultry or wild birds, information regarding this virus will be disclosed in the footnotes. ^c^ *: site of cleavage between HA1 and HA2. ^d^ N.D.: not determined. ^e^ No: no reported progenitor. ^f^ N.A.: not applicable. ^g^ Mass mortality of common terns along the coast of The Cape of South Africa. ^h^ Direct progenitor is unknown, but domestic duck LPAIV A/duck/Victoria/76 (with tribasic cleavage site sequence PEIPKKR*G) has been hypothesized to be the progenitor of this HPAIV. ^i^ Direct progenitor is unknown, but closely related wild bird LPAIVs have been identified (A/tern/Potsdam/342-6/79 and A/swan/Potsdam/63-6/81). ^j^ Reassortant HPAIVs with the subtypes H5N1, H5N2, H5N3, H5N4, H5N5, H5N6, and H5N8 have been detected. ^k^ IVPI of HPAIV isolated from chickens (A/chicken/NSW/1/97). IVPI of emu isolate (A/emu/NSW/97) was 1.3. ^l^ Direct progenitor is unknown, but closely related wild bird LPAIV was detected (A/mallard/Netherlands/12/00). ^m^ HPAIV was retrospectively detected at low frequencies during an LPAIV outbreak in turkeys. ^n^ HPAIV was ostrich-adapted. IVPI increased to 1.2 after one passage in embryonated eggs and to 2.73 after one passage in chickens. ^o^ MBCS-containing AIVs were detected in one healthy whistling duck and multiple spur-winged geese. ^p^ Low IVPI is probably due to ostrich adaptation. ^q^ LPAIV with cleavage site motif PQRKKR*G was detected in 2008 (A/chicken/Taiwan/A703-1/08). ^r^ A total of 1300 dead migratory birds were found in a wetland. ^s^ Direct progenitor is unknown, but HA of LPAIV A/Guizhou/03240/2015 is highest in similarity to that of the early HPAIVs. ^t^ HPAIV was detected in one mute swan that was found dead.

The emergence of HPAIVs is thought to occur in terrestrial poultry following the transmission of H5 or H7 LPAIVs from the wild bird reservoir. The link between HPAIV genesis and terrestrial poultry is evident when an LPAIV progenitor is detected in the same or neighboring poultry farm prior to HPAIV detection. These sparse occasions (12 in total) have been indicated in Table 1. For another four cases, LPAIVs carrying a closely related HA were identified and are mentioned in the footnotes of Table 1. In most cases, a progenitor was not identified and the species in which the LPAIV to HPAIV conversion occurred could not be unambiguously determined. Nevertheless, HPAIV genesis has historically not been associated with wild birds as, prior to the emergence of the Gs/Gd-H5 lineage, HPAIVs had only been sporadically detected in a handful of wild birds near infected poultry populations [102,103] and during one South African outbreak of H5N3 HPAIV in common terns in 1961 [35]. In 1996, the Gs/Gd-H5 HPAIVs were first detected in domestic geese rather than in terrestrial poultry [60]. After becoming enzootic in China and Southeast Asia in 2003 and 2004, they have disseminated to Africa, Europe, and North America, probably by long-distance bird migration [2,104,105,106]. Recently, a few other AIVs with MBCS that could not be linked to outbreaks in poultry holdings have been detected in wild birds. In 2007, H5N2 AIVs with the MBCS motifs KEKRRKKR and REKRRKKR were detected in healthy geese during wild bird surveillance in Nigeria [79], however, the inability to culture the AIVs from original material prevented further investigation into their pathogenicity. In 2016, an H7N1 HPAIV outbreak occurred in a nature reserve in Algeria, leading to a high number of bird deaths [94], and an H7N7 HPAIV was detected in a dead mute swan in Lithuania in 2021 [101]. Taken together, the majority of HPAIV emergences (45 out of 51) could be linked to outbreaks in terrestrial poultry species. However, surveillance gaps and differences in pathogenicity between hosts might lead to the underdetection of the emergence of MBCS-containing AIVs.

## 3. HA Cleavage and Virulence

The HA glycoprotein mediates two essential events during viral entry, i.e., binding to sialic acid moieties on target cells and fusion between the viral and endosomal membranes. HA is produced as precursor HA0, anchored in the membrane of the endoplasmic reticulum [107]. HA0 monomers consist of two disulfide-linked subunits, whose boundaries are marked by a proteolytic cleavage site: subunit HA1 harbors the globular head domain with the receptor-binding pocket, and subunit HA2 harbors a large part of the HA stalk region (Figure 2A,B). Upon translation, HA0 monomers non-covalently assemble as homotrimers. The cleavage site is situated in a membrane-proximal loop (Figure 2B) [108]. The cleavage of HA0 into the HA1 and HA2 subunits activates the fusogenic properties of HA by exposing the first 11 residues of the HA2 N-terminus, which form the hydrophobic fusion peptide (Figure 2A) [109]. The fusion peptide relocates to the interior of the trimer interface upon HA0 cleavage [108]. This conformational change is the prerequisite for membrane fusion, which is triggered in an acidic environment such as the late endosomal compartment upon endocytosis [110,111].

The cleavage of HA0 is performed by host proteases as AIV genomes do not encode proteases. The cleavage site of LPAIVs contains one basic amino acid, an arginine (R) or in some cases a lysine (K). This monobasic cleavage site is recognized by trypsin-like serine proteases that are expressed in a tissue-specific manner. In chickens, trypsin-like proteases are expressed in the RT and GIT, limiting the tropism of LPAIVs to these organ systems. In contrast, the MBCS, here referred to as cleavage site motifs that contain a minimum of four basic amino acids or at least one inserted basic amino acid, from HPAIVs is cleaved by proteases from the subtilisin-like proprotein convertase family, including furin [112,113]. In chickens, the ubiquitous expression of this family of proteases allows for systemic viral dissemination, leading to the severe disease that is associated with HPAIV infections. The removal of the MBCS from HPAIV HAs by reverse genetics confirmed that the MBCS is the main virulence determinant of HPAIVs in terrestrial poultry [114].

The virulence of HPAIVs is a polygenic trait that is, apart from the presence of an MBCS, mediated by other properties of HA as well as other viral proteins. This is exemplified by the fact that the virulence in chickens of an LPAIV H5 with artificially inserted MBCS does not necessarily equal that of a Gs/Gd-H5 HPAIV [115]. Many amino acid changes exist between HPAIVs and their direct LPAIV progenitors, mostly in HA and the polymerase proteins, but no genetic markers common to HPAIVs apart from the MBCS have been identified [116,117]. A large phylogenetic and statistical screening was performed in which HPAIV and LPAIV sequences from different monophyletic lineages were compared, leading to the identification of a subset of parallel substitutions in HA and in the polymerase genes, which are positively associated with the evolution of HPAIV [118]. It is not yet elucidated whether substitutions accompanying HPAIV emergence are neutral, permissive, compensatory, or the result of adaptation to replication in terrestrial poultry. Indeed, AIVs show a high degree of species adaptation, and consequently, adaptations to terrestrial poultry are present in HPAIVs, such as deletions in NA [119,120,121,122,123,124] and NS1 [125,126]. Furthermore, poultry AIVs often have additional glycosylation sites (GSs) in HA that influence replication [120,124,127]. Many other factors that influence AIV virulence have been described, such as the pH stability of HA [128,129,130].

## 4. Proteases That Activate HA

The cleavage of HA0 can occur at multiple stages during the viral replication cycle. HA0 can be cleaved upon translation in infected cells by proteases in the secretory pathway, in the extracellular space by soluble proteases, or on the plasma membrane of target cells following budding or prior to endocytosis [131]. The nature of the particular protease that cleaves HA0 is dependent on its tissue-specific expression and substrate specificity. Furthermore, substrate cleavage efficiencies by various proteases differ between the HA subtypes [132], which might be due to differences in cleavage site sequence composition. Many proteases have been identified as likely candidates to cleave monobasic HA0 (reviewed in [131]). The Type II Transmembrane Serine Protease (TMPRSS2) has been shown to activate most, but not all, human and avian HAs in mice [133,134,135,136,137], human respiratory cell culture models [138,139], and overexpression systems [132]. Human Airway Trypsin (HAT) is expressed in respiratory ciliated cells and can cleave LPAIV HA0, albeit to a lesser extent than TMPRSS2 [132]. This might be due to the fact that HAT is active only at the plasma membrane, whereas TMPRSS2 exerts its activity both at the plasma membrane and in the secretory pathway [140], or due to the substrate specificity differences between TMPRSS2 and HAT [141]. Other proteases that can cleave monobasic HA0 are matriptase [142,143,144] and TMPRSS4 [145]. In rats, the presence of excreted HA0-cleaving proteases has been reported, such as tryptase Clara [146] and mini-plasmin [147]. Little is known regarding the avian counterparts of the mammalian LPAIV-activating proteases. Gotoh et al. found that a blood clotting factor Xa-like protease from embryonated chicken eggs activates LPAIV HA [148]. Recently, it has been shown that mallard duck TMPRSS2 can cleave H1–12, H15, and H16, but not the H14 prototype that was used in the study [139]. The NA of a few specific viral strains can also mediate the non-canonical cleavage of HA0. The NA of the mouse-adapted H1N1 A/WSN/33 strain sequesters plasminogen near HA, which, upon activation by NA, can cleave HA0 [149,150]. Similarly, the NA of LPAIV H7N6 A/mallard/Korea/6L/07 activates prothrombin into thrombin, which cleaves the consensus sequence GR*G (* represents the site of cleavage), which is present in Eurasian H7 isolates [151]. Both NA-mediated HA cleavage mechanisms allow for viral replication outside of the RT and GIT, because plasminogen and prothrombin are present systemically.

Furin was the first protease to be appointed as the activator of HPAIV HA0. Furin belongs to the proprotein convertase (PC) family, consisting of nine proteases that activate protein precursors during post-translational processing. PCs are membrane-attached or soluble and each have specific subcellular residencies (reviewed in [152]). Furin and furin-like PCs (i.e., PC5/6, PACE4, and PC7) are often implicated in the glycoprotein processing of many different viruses due to their ubiquitous/broad expression patterns [152,153]. Apart from furin, PC5/6 has been shown to cleave MBCS motifs, whereas PACE4 and PC7 did not or did so to a lesser extent [154,155]. The membrane-anchored furin and PC5 isoform PC5B accumulate in the Trans-Golgi Network (TGN), but both are present along the secretory pathway [152]. HPAIV HA trimers are therefore cleaved during post-translational processing, mainly in the TGN, which results in the release of infectious virions from the cell [156]. The minimal consensus motif for furin is RXXR, but the motif RXR/KR is considered the minimal furin-cleavage consensus for the cleavage of HA, with X representing any amino acid [152,157]. The MBCS motif KKKR is poorly cleaved by furin but is efficiently cleaved by the ubiquitously expressed Mosaic Serine Protease Large-form (MSPL) and its splice variant TMPRSS13 [158]. Most studies on the role of furin-like proteases in HA cleavage have been performed in mammalian systems. The only data available so far on avian furin-like proteases in relation to HA cleavage is that chicken furin can also cleave HPAIV HA [159], and more work is necessary to increase our knowledge of avian counterparts.

## 5. MBCS Amino Acid Composition

The MBCS amino acid composition, surrounding sequence, three-dimensional loop structure, and accessibility of the cleavage site strongly influence HPAIV H0 cleavage by host proteases. The cleavage site motifs of documented newly emerged HPAIV strains vary extensively in length and in amino acid composition (Table 1; Figure 3). Moreover, MBCS sequences can vary following the initial conversion event, exemplified by the diversity of the MBCS motifs in the Gs/Gd-H5 lineage (reviewed in [160]). Once an MBCS has been acquired in HA, variation at the nucleotide and amino acid level due to subsequent substitutions and/or insertions/deletions, which might be favored by the presence of purine-rich sequences in HPAIV MBCS motifs, might be tolerated as long as the resulting cleavage site can be cleaved by furin-like proteases. Although the basic amino acid arginine can be coded for by six codons, MBCS motifs almost exclusively contain purine-rich arginine codons (AGA and AGG) (Figure 3). The minimal consensus sequence of AIV-cleaving furin-like proteases is RXR/KR, but the amino acid sequence up to position six relative to the C-terminus of HA1 (P6) is thought to influence furin cleavage [161,162]. Mutational studies on the P1 to P4 positions in HPAIV HAs determined that altering either the P1 or P4 arginines resulted in reduced cleavability, whereas the P2 and P3 positions allowed for more flexibility [157,163]. Most H5 and all H7 MBCS motifs contain amino acid insertions of basic and/or non-basic amino acids in the cleavage site region (Figure 3). Two-amino-acid insertions are most frequently observed amongst natural MBCS motifs and the longest reported insertion is of 20 amino acids in a laboratory H7N7 strain [164]. It is hypothesized that amino acid insertions enlarge the cleavage loop so that it protrudes into the solvent, making it more accessible to proteases. Some H5 HPAIVs harbor a minimal furin cleavage site (Figure 3A; Table 2), but the presence of insertions leads to more optimal furin cleavage [163], and upon the passage of viruses containing cleavage site motifs RKTR and RKKR in chickens, HPAIVs with insertions at the HA cleavage site had a selective advantage [165]. In contrast, insertions were shown to be essential for the trypsin independence of an H7N2 AIV, as just substitutions of non-basic by basic amino acids in the LPAIV variant were insufficient, even when the resulting cleavage site motif was PEKRKKR [166]. In some HPAIVs, the cleavage site sequence deviates from the RXR/KR consensus motif, such as the MSPL-cleaved H5 HAs as well as H7 HAs originating from NHR (Figure 3B). Following NHR, MBCS motifs often contain RX^nb^X^nb^R, where X^nb^ is any non-basic amino acid. Such a motif can still be cleaved by furin, as was demonstrated for a laboratory H7N3 AIV with a large insertion, resulting in the MBCS motif SLSPLYPGRTTVLHVRTAR [167]. H7 MBCS motifs that arose through NHR also contain more histidines than MBCS motifs that arose through other mechanisms, although not in the positions essential for furin cleavage. The presence of basic histidine residues in the MBCS has been shown to contribute to furin cleavage, i.e., for the MBCS motif NSTHKQLTHHMRKKR in an equine H7N7 strain [168].

The presence of an MBCS in AIVs does not automatically confer trypsin-independency or high virulence in vivo [169]. As indicated in Table 2, multiple MBCS-containing AIVs with a tetrabasic MBCS have been classified as LPAIV due to an IVPI below 1.2. A well-documented example is that of the H5 AIV that caused an outbreak in Pennsylvania in the USA in 1983. AIV isolates from the early stages of the outbreak harbored the KKKR cleavage site motif, but had low virulence in poultry (IVPI: 0). AIV isolates from later stages of the outbreak gained virulence (IVPI: 2.37), but no changes in the MBCS motif were observed [47]. Kawaoka et al. attributed the change in virulence to the loss of an N-linked glycosylation site at position 22 in HA (H3 numbering) [170]. This carbohydrate neighbors the cleavage loop, therefore causing steric hindrance and blocking protease access to the cleavage site (Figure 2B). An artificial increase in the number of basic amino acids can overcome the hindering presence of the carbohydrate chain [73,171,172]. The loss of this putative glycosylation site at p22 has been reported as a mechanism for virulence gain in AIVs of the H5 and H9 subtypes [171,173] and its impact is HA-dependent. Nevertheless, the presence of the putative glycosylation site p22 does not always hinder the cleavage of tetrabasic MBCS motifs as multiple H5 HPAIVs containing that combination have been described (Table 2). Conversely, a trypsin-dependent H5 AIV from Texas (IVPI: 0), harboring an RKKR MBCS and glycosylation p22, did not become trypsin-independent upon removal of the glycosylation site p22 [73].

**Table 2 viruses-14-01566-t002:** Natural occurrences of H5 AIVs containing a tetrabasic MBCS: molecular and in vivo characteristics.

Viral Strain	Cleavage Site Sequence	Trypsin-Independent HA Cleavage	Putative GS p22 ^a^	IVPI ^b^ (Phenotype ^c^)	Ref.
A/chicken/Scotland/1959 (H5N1)	PQRKKR*G ^d^	+	-	N.D. ^e^ (HP)	[171]
A/chicken/Pennsylvania/1/1983 (H5N2)	PQKKKR*G	-	+	0 (LP)	[171,172]
A/chicken/Pennsylvania/1370/1983 (H5N2)	PQKKKR*G	+	-	2.37 (HP)	[171,172]
A/chicken/Texas/298313/2004 (H5N2)	PQRKKR*G	-	+	0 (LP)	[73]
A/chicken/Taiwan/A703-1/2008 (H5N2)	PQRKKR*G	+	+	0.89 (LP)	[174,175]
H5N2 HPAIV from Taiwan 2012 ^f^	PQRRKR*G	N.D.	+	2.91 (HP)	[176]
A/chicken/France/150169a/2015 (H5N1)	HQRRKR*G	N.D.	+	2.9 (HP)	[93]

^a^ Presence of putative glycosylation site (GS) p22 (H3 numbering), based on amino acid sequence only. ^b^ IVPI: intravenous pathogenicity index. ^c^ Phenotype of viral strain is based on whether the birds were showing severe (HP) or no (LP) symptoms during the time of detection and/or IVPI. ^d^ *: site of cleavage between HA1 and HA2. ^e^ N.D.: not determined. ^f^ It is unclear which exact strain was used for IVPI determination.

## 6. Subtype Restriction of MBCS Acquisition: Compatibility of an MBCS at the Protein Level

All naturally evolved AIVs with an MBCS, defined as a tetrabasic motif or with inserted basic amino acids, are of the H5 and H7 subtypes, except for the previously mentioned trypsin-dependent H4N2 isolate with cleavage site motif PEKRRTR (IVPI: 0) [17,177]. However, non-H5/H7 LPAIVs with a tribasic cleavage site due to substitutions (within P1 to P4) have been described, e.g., canine H3N2 virus (PERRTR) [178] and many H9N2 LPAIVs (PAKSKR, PARSRR, and PARSKR) [179,180,181]. Why the acquisition of an MBCS is restricted to the AIVs from the H5 and H7 subtypes has not yet been elucidated, but multifactorial hypotheses have been proposed.

Firstly, it was hypothesized that the putative glycosylation site p22, which can reduce HA cleavage efficiency by furin-like proteases, might be less present in H5 and H7 AIVs [182]. However, it is conserved in AIVs from all 16 avian HA subtypes, including H5 and H7 [182] and the highest frequencies of LPAIVs without putative glycosylation site p22 have been detected in non-H5/H7 subtypes (in 3.2% (242/7657) of H9 sequences) [182]. Yet, this absence can potentiate trypsin-independent HA cleavage in certain H9N2 LPAIVs [179]. Secondly, the subtype specificity of HPAIV emergence is not due to the fact that H5 and H7 LPAIVs circulate to higher extents in poultry populations, in which virtually all MBCS acquisitions have been detected, than the other LPAIVs. H6, H9, and H10 LPAIVs are also frequently detected in poultry [183,184].

Thirdly, it has been hypothesized that the subtype restriction of MBCS acquisition might be the result of incompatibility at the protein level of an MBCS in HAs from subtypes other than H5 and H7. Nevertheless, reverse genetics studies have shown that HAs from non-H5/H7 subtypes can accommodate an artificial MBCS, which can be cleaved by ubiquitous proteases [185,186,187,188,189,190,191]. However, trypsin-independent MBCS-containing non-H5/H7 AIVs are not necessarily highly pathogenic in chickens, as both the HA and the internal gene cassette contribute to the highly pathogenic phenotype [115,177,185,186]. An H3N8-MBCS AIV did not result in severe disease upon the oculonasal inoculation of chickens [190], whereas an H6N1-MBCS AIV did show systemic replication upon the intranasal inoculation of chickens (IVPI: 1.4) [187]. Veits et al. characterized AIVs that contained H1, H2, H3, H4, H6, H8, H10, H11, H14, or H15 HAs with an engineered MBCS and the remaining seven segments of an H9N2 LPAIV or H5N1 HPAIV [185]. All reassortants had HPAIV characteristics in vitro, but only the H2, H4, H8, and H14 HAs in an HPAIV H5N1 genetic background resulted in high virulence in vivo. Interestingly, Gischke et al. reported the in vivo attenuation of an H4N2 AIV with a PEKRRTR cleavage site when the T was mutated to R or K, despite a modest increase in trypsin-independent activation by endogenous proteases [177]. Furthermore, the H4 virus with PEKRRKR cleavage site motif only supported an HPAIV phenotype when combined with the internal genes of an H5 HPAIV [177]. The virulence of MBCS-containing LPAIVs can increase upon consecutive passaging in chickens, as was exemplified by a trypsin-dependent H9N2-MBCS AIV (PARKKR) that acquired trypsin independence (following the removal of GS p22), associated with increased morbidity and mortality upon intravenous inoculation, over the course of ten passages [173]. To conclude, AIVs from non-H5/H7 subtypes with artificial MBCS show an HPAIV phenotype in vitro, which indicates that there are no major structural constraints in the HA protein, explaining the absence of MBCS motifs in non-H5/H7 AIVs. The restriction of HPAIV to H5 and H7 subtypes is, therefore, most likely due to differences and/or constraints at the RNA level, which will be discussed later in this review. However, the in vivo pathogenicity of MBCS-containing non-H5/H7 AIVs can remain low due to virulence determinants beyond the MBCS or suboptimal HA cleavage, which might hamper natural selection. Furthermore, the lack of LPAIV adaptation to poultry might partially explain why some MBCS-containing AIVs do not display a full HPAIV phenotype in vivo, especially because many of the LPAIVs in the above-mentioned studies were isolated from wild birds.

## 7. Mechanisms of MBCS Acquisition

Although the exact mechanisms by which H5 and H7 LPAIVs acquire an MBCS have not been elucidated, multiple non-exclusive hypotheses have been proposed [192,193,194,195]. The hypotheses are based on sequence analyses of MBCS motifs from naturally occurring HPAIVs and consist of (A) the substitution of single nucleotides resulting in codons coding for R or K, (B) stuttering or backtracking on homopolymer-rich and/or realignment-prone sequences resulting in duplications, and (C) NHR between HA RNA and RNA from viral or host origin. Both H5 and H7 HPAIVs are thought to have been generated through nucleotide substitutions and/or backtracking (Figure 3A,C), whereas NHR has been so far restricted to H7 HPAIVs (Figure 3B). These MBCS acquisition mechanisms could occur in a stepwise manner [196,197]. The following paragraphs will discuss each of the hypothesized mechanisms and the related factors that set H5/H7 HA apart from HA from other subtypes.

### 7.1. Nucleotide Substitutions

The influenza A virus RNA-dependent RNA polymerase (RdRp) is inherently error-prone. Single nucleotide substitutions occur at an approximate rate of 2.5 × 10^−5^ substitutions per nucleotide per replication cycle [198]. The consensus sequence of the HA cleavage site of H5 LPAIVs is RETR, with glutamic acid and threonine encoded by GAA and ACA codons, respectively [183]. Both codons can be converted into a basic amino acid-encoding codon upon only one nucleotide substitution, e.g., into AAA (K) or AGA (R). Such substitutions resulted in the genesis of H5 HPAIVs containing tetrabasic MBCS motifs, such as A/chicken/Scotland/1959 (RKKR) and A/chicken/France/150169a/2015 (RRKR) (Table 2). Low frequencies of amino acid substitutions within H5 and H7 LPAIV cleavage sites, resulting in tribasic motifs, have been detected [160,199].

### 7.2. Influenza Virus RdRp Stuttering and Backtracking

MBCS insertions are often duplications of neighboring sequences, as can be appreciated by comparing HPAIV and LPAIV cleavage site sequences (Figure 3A,C). Such duplications are thought to arise through the backtracking and realignment of the RdRp. This probably occurred in an HPAIV from the Mexican H5N2 outbreak in 1994/1995, during which an LPAIV is thought to have acquired an MBCS by one substitution followed by the duplication of six nucleotides (Figure 3A) [200]. The duplication seems to have occurred twice, eventually resulting in an RKRKRKTR cleavage site. The stuttering of the RdRp, i.e., repeatedly replicating the same nucleotide in a homopolymer stretch, could explain the presence of long adenine (A; in cRNA orientation) stretches often observed in MBCS sequences (Figure 3A,C). Stuttering on uracil residues has been well-described for poly-A-tail formation in mRNA by the viral RdRp, but this is dependent on physical constraints due to the 5′ vRNA hook remaining bound to the RdRp [201], which is not applicable to the MBCS region.

#### 7.2.1. Influence of the Nucleotide Sequence on Influenza Virus RdRp Stuttering and Backtracking

Templates with adenine/uracil stretches have been shown to be more prone to influenza virus RdRp slippage than those with cytosine and guanine stretches, probably due to weak interactions between the base pairs [202]. Interestingly, A-rich codons often code for basic amino acids R (AGA amongst others) and K (AAA and AAG). Therefore, duplication or stuttering on A-rich templates often result in the insertion of basic amino acids at the protein level. It has been hypothesized that H5 AIVs are prone to RdRp slippage due to high numbers of adenines, and purines in general, at the cleavage site [200]. Nao et al. compared the average number of purines at the cleavage site region between a subset of HA subtypes and found that H5 and H7 HA sequences isolated in ducks contained more purine-rich sequences than H4, H6, H9, and H16 [203]. We have performed a comprehensive analysis investigating the average number of adenines and purines at the cleavage site region in all publicly available avian LPAIV HA sequences from all subtypes [183]. Large differences between HA subtypes regarding adenine usage in cRNA were detected. H5 cRNA sequences stood out as particularly A-rich, with 85% of sequences having eight or more As in P1 to P4, not necessarily consecutive. Other A-rich sequences were observed in the H3 and H14 subtypes, with 27% and 70% of sequences having eight or more As, although the latter might be unreliable due to the low number (33) of available sequences. In contrast, H7 LPAIVs contained a number of As close to average. The number of consecutive As within A-stretches per HA subtype was also analyzed, but in this regard, H5 and H7 HAs did not stand out from the other subtypes [183]. On the other hand, when considering purine or pyrimidine nucleotides, both H5 and Afro-Eurasia-Oceania (AEO) H7 sequences stand out from other subtypes, with the highest number of purines or the longest purine stretch length in the P1 to P4 region, respectively. These results mark H5 and AEO H7 LPAIVs as possibly more prone to RdRp stuttering and backtracking than other subtypes due to a higher adenine/purine content in the cleavage site.

These hypotheses are corroborated by results from experiments on the genetic stability of different sequences at the HA cleavage site. The serial passaging of LPAIVs with mono-, di-, or tribasic cleavage sites in cell culture without trypsin, embryonated eggs, or chickens to generate and/or select for HPAIVs has been attempted (Appendix A) [164,165,172,173,175,196,204,205,206,207,208,209,210,211,212,213,214,215,216]. Many of these experiments were performed using field isolates rather than clonal recombinant viruses, preventing the differentiation between MBCS genesis and the selection of already present minor HPAIV populations. Furthermore, multiple studies have applied next-generation sequencing to assess the presence of MBCS motifs at low frequencies [197,203,210,212]. However, the results of these studies should be interpreted with care, as homopolymer stretches are intrinsically error-prone for the replication enzymes used during the sequencing procedure. Nevertheless, it is apparent from passaging experiments that MBCS acquisition and HPAIV emergence are difficult to reproduce in experimental settings and require stringent selection pressure. Strains harboring the H5 LPAIV consensus RETR motif show high genetic stability during passaging [165,210]. For example, Ito et al. passaged an H5N3-RETR isolate 24 times in chicken air sacs, during which T was substituted for K, and five times in the brain, during which E was substituted for R and an R was inserted, resulting in the motif RRKKR [196]. In contrast, upon the inoculation of chickens or embryonated eggs with LPAIVs containing a dibasic cleavage site, and consequently containing longer stretches of adenines (cRNA), additional basic residues were readily introduced [165,211]. The genetic instability of di- and tribasic cleavage sites has also been shown previously using non-clonal isolates as starting material [205,206]. Increased nucleotide insertion rates were observed in in vitro reporter assays when the template contained long stretches of uracils [180,203,212] and insertions were acquired rapidly when clonal AIV stocks that already contained an MBCS were passaged once in chickens [165,173,212]. However, not all MBCS motifs are genetically unstable, e.g., H5N2-RKKR [175] and H9N2-RKKR [173], whose cleavage sites contain homopolymer stretches of seven and six adenines, respectively, and yet remained stable upon passaging (respectively eight and ten times) in chickens.

The results from the abovementioned passaging experiments and in vitro reporter systems [180,203,212] suggest that the initial acquisition of additional adenine (cRNA) nucleotides by substitutions might be the bottleneck for MBCS acquisition, as the resulting homopolymer-rich cleavage site sequence is more prone to subsequent insertions. We therefore hypothesize that H5 and H7 HAs are more prone to acquire an insertion-prone minimal MBCS than those from other subtypes. This could be due to a combination of nucleotide usage in H5 and H7 LPAIVs HAs and/or the positive selection of variants with a minimal MBCS that would be better cleaved by proteases and therefore have a fitness advantage. We investigated whether H5 and H7 LPAIV Has are more prone to acquire an MBCS due to the specifics of their cleavage site sequences, i.e., sequence differences and codon usage [183]. To that end, we assessed how many nucleotide substitutions are required in LPAIV cleavage site sequences in order to form a cleavage site motif that acts as a stepping stone for further mutational events. Such a motif was defined as containing at least three basic amino acids in P1 to P4 and an arginine in P1, excluding histidine and pyrimidine-containing arginine codons based on codon usage in HPAIVs [183]. Strikingly, an average of two to three substitutions were required in HAs from most subtypes to obtain a tribasic cleavage site (Figure 4). In contrast, the H3, H5, H7, and H9 subtypes contained a high percentage of HA sequences for which only one substitution was required to form a potentially insertion-prone tribasic cleavage site. However, the P3 serine in H9 was the remaining non-basic amino acid in such a tribasic cleavage site, coded for by TCA or TCT, interrupting the stretch of adenines which might disfavor insertions [180]. There was also an unfavorable ACC threonine codon present in the resulting H3 tribasic cleavage sites. In addition, RNA stem-loop structures, which have been suggested to potentiate stuttering and backtracking, as will be discussed in Section 7.2.2, are not conserved in the majority of avian H3 lineages [194], potentially explaining why H3 HAs have so far not acquired an MBCS in nature. Interestingly, fewer substitutions, on average, were required in AEO H7 LPAIVs to form a non-consecutive tribasic cleavage site than in those from the Americas—one and two, respectively [183]. It is tempting to speculate that this difference contributes to the observation that American H7 MBCS acquisition primarily involves NHR, whereas insertions via backtracking/stuttering are common in AEO H7 HPAIVs (Figure 3). Furthermore, when the number of required substitutions to mutate to a tribasic cleavage site were subdivided over three species categories (i.e., Anseriformes and Charadriiformes, terrestrial poultry, and others), it was apparent that fewer substitutions were required in H5 HAs from viruses detected in terrestrial poultry than in those from viruses from the other species [183]. This might indicate that there is more selection pressure on cleavage site sequences containing multiple basic amino acids in terrestrial poultry than in other birds, such as wild waterfowl. The abovementioned analysis does not take codon usage at P1 to P4 into account, thus a second analysis was performed that only allowed for codons that were naturally detected on each position per HA subtype, but the number of required substitutions to reach a stretch of a predetermined number of As was determined [183]. H5 and H7 HAs required the fewest mutations to reach long A-stretches, followed by H6 HAs. Taken together, this analysis indicates that LPAIVs from the H5 and H7 subtypes might have a genetic predisposition for acquiring an MBCS because of their cleavage site nucleotide sequence, which requires fewer random substitutions than HAs from other subtypes to obtain a cleavage site sequence that is prone to insertions.

#### 7.2.2. Influence of RNA Structure on Influenza Virus RdRp Stuttering and Backtracking

RNA folding in the region coding for the HA cleavage site has been suggested to influence RdRp backtracking/stuttering [55,70,91,180,194,197,200,203]. Although viral RNA is bound by NP molecules, local RNA structures are still formed due to the non-uniformity of NP binding [217,218]. Subtype-specific conserved stem-loop structures in the region coding for the cleavage site have been predicted by RNA folding algorithms and covariation analyses (Figure 5A) [194,219]. In H5 and H7 HA, additional basic amino acid-coding codons are inserted in the predicted loop, increasing its size [91,194,203]. Nevertheless, the presence of conserved secondary structures at the cleavage site is not unique to H5/H7 HAs, as similar stem-loop structures have been identified in non-H5/H7 HAs, albeit not in all subtypes [194,220]. Therefore, the conserved RNA secondary structures at the HA cleavage site might have a role beyond MBCS acquisition, such as the stalling of ribosomes during translation to allow the correct folding of the HA1/HA2 connecting peptide. An extensive phylogenetic and structural study, comparing LPAIV lineages that never gave rise to an HPAIV and lineages from which an HPAIV emerged, hinted at the presence of specific RNA structures in American H7 LPAIVs associated with MBCS acquisition via NHR [220]. The thermodynamic stability of these RNA secondary structures did not correlate with the capacity to become HPAIV [194,219,220]. We have previously hypothesized that the RdRp can get trapped while replicating RNA that contains strongly paired stem structures, promoting stuttering and backtracking [194]. The RdRp template entrance and exit channel are in close proximity [221], suggesting that parts of the template entering and exiting the RdRp could interact during replication. This would allow the local refolding of the HA stem, stalling the RdRp in the loop that is threaded through the polymerase, forcing it to stutter/backtrack and insert non-templated nucleotides in adenine/uracil-rich loop sequences (Figure 5B). Detailed functional studies using structured templates of HAs from different subtypes will hopefully determine whether cryptic differences between RNA stem-loop structures, such as local RNA folding during cRNA or vRNA replication, could explain subtype-specific MBCS acquisition. The first studies applying such a template-based approach in in vitro reporter systems hint toward the importance of the presence of the structure and size of the loop for MBCS acquisition and note that H5 LPAIVs contain larger loops than LPAIVs from other subtypes [180,203,212].

### 7.3. Non-Homologous Recombination

The insertion of exogenous RNA into the cleavage site sequence following NHR has been shown in seven out of 33 natural H7 HPAIV emergences and is probable in four additional H7 emergences based on codon usage and sequence alignment, as the stretch of inserted nucleotides did not consist of duplications of the surrounding sequence (Figure 3B,C; Appendix A) [195,222]. NHR has also occurred on at least two separate occasions upon the passaging of H7 viruses in experimental settings [164,214]. NHR cases are considered to be confirmed when the origin or the inserted RNA can be reliably established by blast. Most of the NHR cases belong to the American H7 lineage (Figure 3B), whereas insertions via backtracking/stuttering are thought to have occurred primarily in Eurasian H7 HPAIVs (Figure 3C). Inserted RNA either originated from host 28S ribosomal RNA (rRNA) [85,98,195,214], or viral RNA such as the nucleoprotein (NP) gene [65,164] or matrix (M) gene [74]. Some of the shorter inserted RNA sequences have been suggested to be derived from transfer RNA [195]. An equine H7N7 virus contained a stretch of 10 inserted amino acids in the cleavage site, but subsequent substitutions in this sequence prevented the determination of the original source RNA [168]. Surprisingly, recombination in H5 HA has been detected by the next-generation sequencing of samples from the H5N2 virus serially passaged in ovo [210]. Moreover, the MBCS from the 1961 H5N3 outbreak in South Africa might have arisen through NHR, as the cleavage site contains codons that are unlikely to have arisen through the duplication of neighboring sequences (Appendix A). Little is known of the mechanism that underlies the recombination of HA RNA with exogenous RNA. The presence of (partially) palindromic sequences flanking some cleavage sites has been suggested to play a role in some cases, but the significance of this is unclear [85,214]. We recently showed that virtually all sequences of exogenous RNA inserted in H7 correspond to small nucleolar (sno) RNA binding sites. SnoRNAs are cellular RNAs, whose main function is to guide RNA modifying proteins (e.g., methyltransferase) through complementary binding sites in the rRNA [223]. Interestingly, we found that two snoRNA binding sites out of more than 100 in 28S rRNA were involved in five independent recombination events with rRNA. Additionally, we identified chicken snoRNA binding sites corresponding to the known sites of the recombination of H7 with NP and M segments. Based on these observations, we hypothesized that snoRNAs facilitate the recombination of H7 HA with exogenous RNA [195], but further studies are warranted to prove or disprove this hypothesis.

## 8. Concluding Remarks

There have been a total of 51 independent emergences of MBCS-containing AIVs in the past 60 years, and their frequency has increased over the past decades. The importance of an MBCS as a virulence determinant was already established decades ago. More recently, knowledge has been acquired about factors beyond the MBCS that determine the pathogenicity of AIVs in avian hosts. Interesting discoveries were made recently regarding HA-activating proteases, such as the systemic dissemination of LPAIVs due to alternative HA cleavage mechanisms, and work is underway to characterize proteases in avian species. The biggest remaining questions concern the emergence of HPAIVs at the molecular level and the restriction of the HPAIV phenotype to the H5 and H7 subtypes in nature. Here, we shed light on the different aspects that might explain the subtype restriction of HPAIV emergence. Viruses with HAs of the H5 and H7 subtypes do not stand out from those of other subtypes with regards to circulation in terrestrial poultry, the glycosylation of p22, the presence of RNA structures at the cleavage site region, or the intrinsic ability to accommodate an MBCS at the protein level. The presence of yet-undetermined factors within H5 and H7 HA might influence the genesis of HPAIVs. Our recent analysis confirmed the previously published notion that H5 LPAIVs harbor exceptionally high numbers of adenine and purine residues at the cleavage site, which might promote errors by the RdRp. Furthermore, we showed that the minimum number of substitutions necessary for an LPAIV to obtain an insertion-prone cleavage site sequence is significantly lower in H5 and H7 subtypes. We postulated that the chance of H5 and H7 HAs acquiring an MBCS in this multi-step process is therefore higher. Recent advances have been made by using reporter assays and deep sequencing methods to elucidate the mechanisms of MBCS acquisition, highlighting that the RNA template sequence seems to be crucial for RdRp stuttering and backtracking. The role of local RNA structures remains to be investigated. Future studies utilizing in vitro replication systems and ultra-sensitive deep sequencing methods will provide the reproducibility, flexibility, throughput, and sensitivity that are necessary to elucidate the specific details of stuttering, backtracking, and recombination by the influenza RdRp.

## Figures and Tables

**Figure 1 viruses-14-01566-f001:**
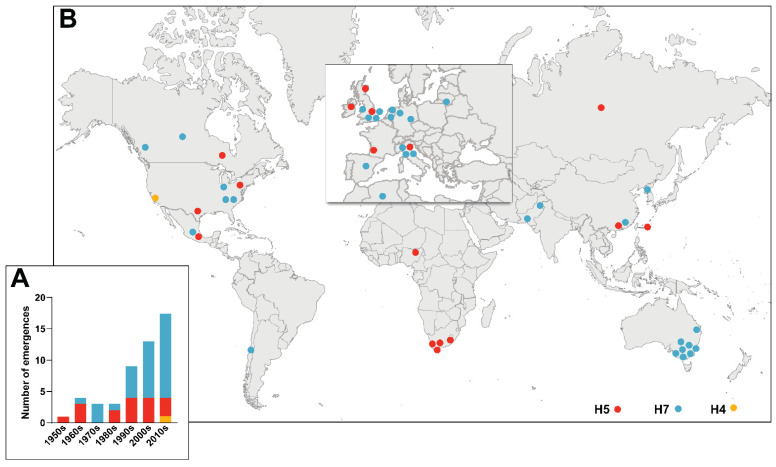
Chronological and geographical depiction of the historical emergence of AIVs with MBCS 1959–2021. (**A**) Number of detections of novel AIVs with MBCS of the H5 (red), H7 (blue), and H4 (orange) HA subtypes per decade. (**B**) Geographical origin of MBCS-containing AIVs. Dots have been slightly displaced in areas with high emergence density in order to improve visibility.

**Figure 2 viruses-14-01566-f002:**
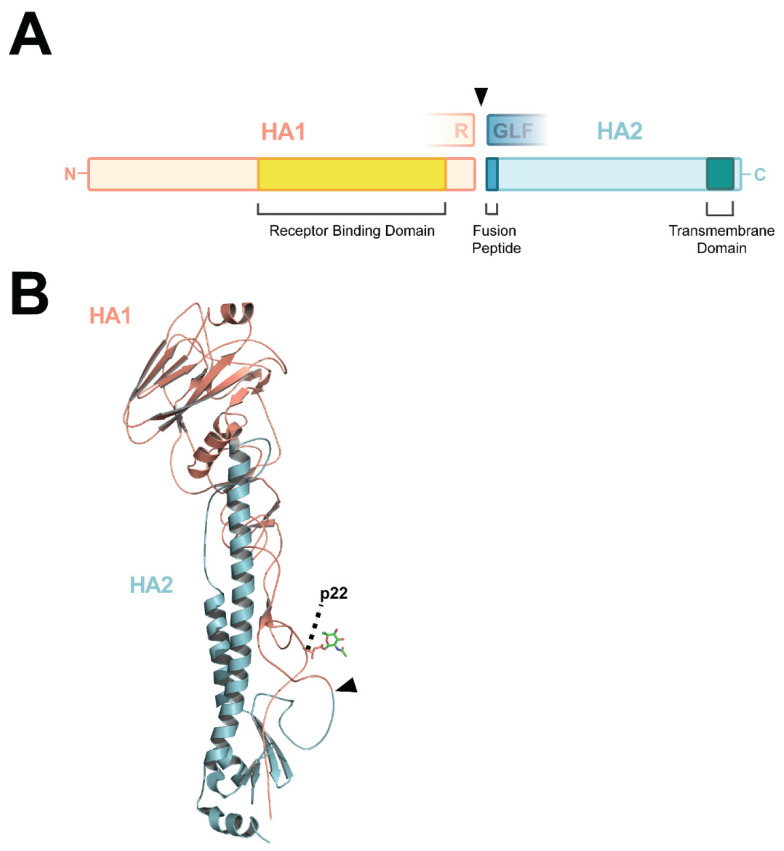
Schematic representation and structure of the HA protein. (**A**) Schematic representation of HA following the cleavage of HA0 into the HA1 and HA2 subunits. The amino acids flanking the cleavage site (arrow head), fusion peptide, and the receptor binding and transmembrane domains are depicted. (**B**) Structure of H3 HA0 monomer (PDB entry 1HA0; non-cleaved R329Q mutant [108]), made in the PyMOL Molecular Graphics System version 2.5.2 Schrödinger LLC, showing the membrane-proximal cleavage loop and site (arrow head), and glycosylation site on position 22.

**Figure 3 viruses-14-01566-f003:**
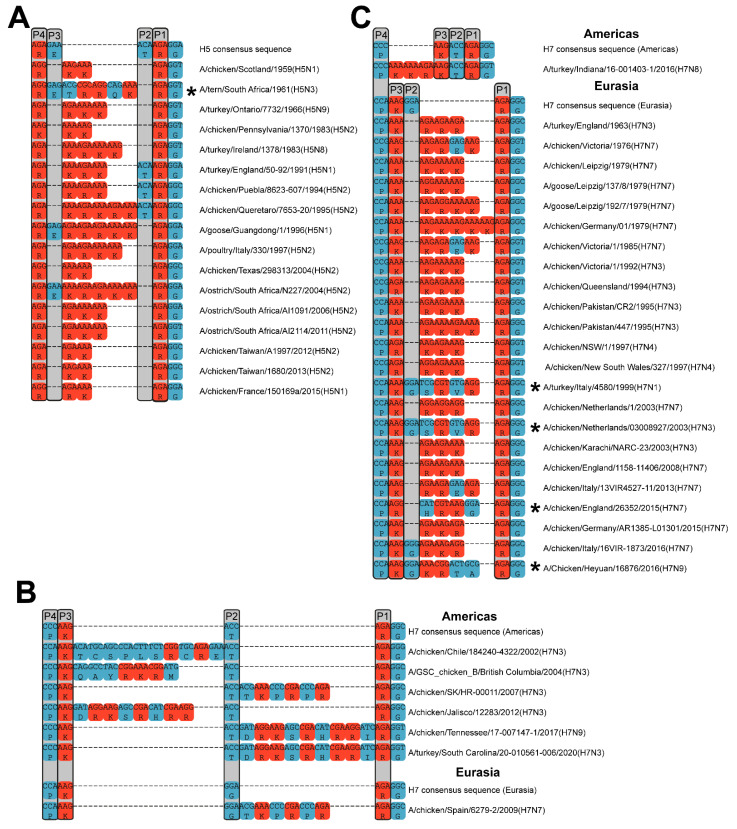
Alignment of cleavage site regions of all available newly emerged natural H5 and H7 MBCS-containing AIV sequences. Nucleotide and amino acid alignments of H5 MBCS-containing AIVs (**A**) and alignments of H7 MBCS-containing AIVs that did (**B**) or presumably did not (**C**) arise through NHR. The asterisks indicate strains that might have emerged through NHR based on codon usage and sequence alignment. The grey boxes delineate conserved amino acids from the LPAI consensus sequence with P1 to P4 indicated on top. Arginines and lysines are depicted in red and all other amino acids are depicted in blue. All sequences are available in fasta format as Appendix A.

**Figure 4 viruses-14-01566-f004:**
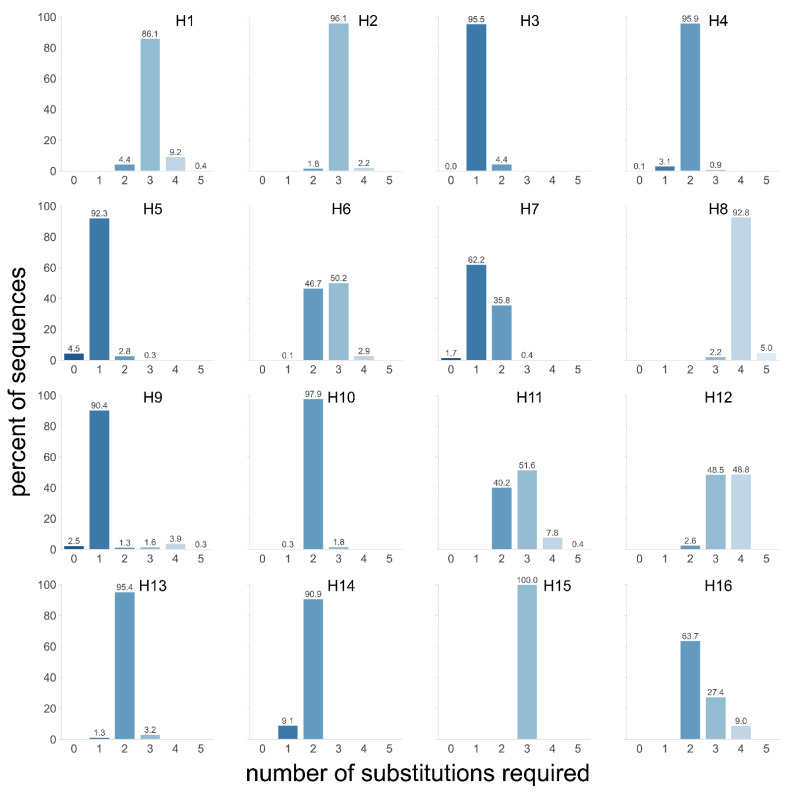
Number of nucleotide substitutions necessary to acquire a tribasic cleavage site in HAs from LPAIVs. The number of single nucleotide substitutions required to obtain a tribasic cleavage site, defined as containing at least three basic amino acids in P1 to P4 and an arginine in P1, excluding histidine and pyrimidine-containing arginine codons based on codon usage in HPAIVs, in all reported LPAIV sequences from H1–H16 HA subtypes. The exact percent of sequences is indicated on top of each bar and a darker blue color indicates fewer substitutions required (Adapted from [183], reproduced with permission from Mathis Funk, Viruses; published by MDPI, 2022).

**Figure 5 viruses-14-01566-f005:**
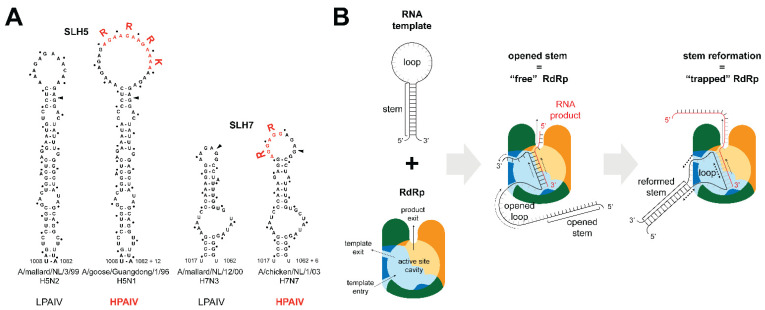
RNA secondary structures in the cleavage site region and their hypothesized influence on RdRp replication. (**A**) Examples of predicted cRNA structures encompassing the cleavage site region in H5 and H7 LPAIV and HPAIV. The predicted stem-loop (SL) structures are conserved in the H5 and H7 lineages. The nucleotides coding for (part of) the MBCS are inserted in the loop and depicted in red. The boundary between HA1 and HA2 is depicted with an arrowhead. Codons are distinguished from each other by dots (Adapted with permission from [219], available under the Creative Commons Attribution 4.0 International License). (**B**) Proposed model for increased stuttering and backtracking rates in the stem-loop region of HA due to RNA structure. The RdRp (PB1 in blue; PB2 in orange; PA in green) replicates the viral genome into product RNA (in red). Due to the close proximity of the template entry and exit channels in PB1, local structures based on complementary sequences in the template can form around the RdRp. The RdRp is trapped in the loop region, resulting in increased rates of stuttering and backtracking, leading to duplications in the RNA product.

## Data Availability

Not applicable.

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
