# Peer review of "Hemagglutinin Subtype Specificity and Mechanisms of Highly Pathogenic Avian Influenza Virus Genesis"

_viruses, 2022, doi:10.3390/v14071566_

Round 1

Reviewer 1 Report

The perfect scientific work and high professionalism.

A small remark:

Words in Latin (in vitro, in vivo, in ovo, Anseriformes and Charadriiformes) should be better to italicize. However, it depends on the rules of the journal. 

Author Response

We thank the reviewers for their constructive critiques. Please find their comments reproduced below, followed up by each of our answers in italics.

Comments and Suggestions for Authors: Reviewer 1

The perfect scientific work and high professionalism.

We thank the reviewer for this very positive assessment.

A small remark:

Words in Latin (in vitro, in vivo, in ovo, Anseriformes and Charadriiformes) should be better to italicize. However, it depends on the rules of the journal. 

Thank you for this comment. We have italicized the bird orders (line 26), but did not change the formatting of other words according to the rules of the journal.

Reviewer 2 Report

The submission by Anja C.M. de Bruin et al., entitled ‘Hemagglutinin Subtype-Specificity and Mechanisms of Highly Pathogenic Avian Influenza Virus Genesis’ is a well compiled review about the genetic mechanisms involved in the evolution of highly pathogenic avian influenza viruses. The manuscript is well compiled and well written. A few minor queries need to be addressed to make the review more perceivable. 

1.   Introduction section:

a.     Description about the classification of HPAIVs (more specifically IVPI) should be described in more detail with references.  

2.     Little light should be thrown on the reassortment events that lead to the genesis of HPAIVS before their mechanism described.

3.     There are many reviews about HPAIV genesis (e.g., Luczo JM et al.,Rev Med Virol. 2015 Nov;25(6):406-30). Authors should accommodate major findings from those articles to make the review more substantial.

4.     Changes in Hemagglutinin Receptor-Binding Specificity also leads to the genesis of HPAIVs. It needs to be discussed.

Author Response

We thank the reviewers for their constructive critiques. Please find their comments reproduced below, followed up by each of our answers in italics.

Comments and Suggestions for Authors: Reviewer 2

The submission by Anja C.M. de Bruin et al., entitled ‘Hemagglutinin Subtype-Specificity and Mechanisms of Highly Pathogenic Avian Influenza Virus Genesis’ is a well compiled review about the genetic mechanisms involved in the evolution of highly pathogenic avian influenza viruses. The manuscript is well compiled and well written. A few minor queries need to be addressed to make the review more perceivable. 

Thank you for the positive assessment.

  1. Introduction section:
  2. Description about the classification of HPAIVs (more specifically IVPI) should be described in more detail with references.  

Thank you for this comment. We have described the IVPI determination in more detail in lines 53-56. This paragraph refers to the official OIE guidelines that are referenced in reference 11 (line 49) and thus we think that additional references are not necessary.

  1. Little light should be thrown on the reassortment events that lead to the genesis of HPAIVS before their mechanism described.

      In the current manuscript, we focus on the genesis of novel HPAIV hemagglutinins by the acquisition of an MBCS, i.e., the conversion from LPAIV to HPAIV. Subsequently, HPAIV HA can be introduced into another genetic background (LPAIV or HPAIV) upon reassortment, leading to the emergence of HPAIV viruses with novel gene constellations. However, these reassortment events do not lead to changes in the HA cleavage site, we therefore think that this aspect is outside the scope of this review. Furthermore, reassortment leading to the genesis of HPAIVs with novel genetic constellations has been reviewed extensively in Dhingra MS et al. Front Vet. Sci. 2018 (PMID:29922681), which is additionally referenced in line 75.

  1. There are many reviews about HPAIV genesis (e.g., Luczo JM et al., Rev Med Virol. 2015 Nov;25(6):406-30). Authors should accommodate major findings from those articles to make the review more substantial.

      We acknowledge that there are many reviews that give an overview of the epidemiology and genesis mechanisms of HPAIVs and we have added references to these reviews in line 75. Furthermore, we have emphasized the reference to Luczo et al. more explicitly in line 274, and it is also referenced in line 420. Furthermore, the review “Genetic changes that accompanied shifts of low pathogenic avian influenza viruses toward higher pathogenicity in poultry” by Abdelwhab et al. is referenced in line 208 and the EFSA report “Mechanisms and risk factors for mutation from low to highly pathogenic avian influenza virus on risk factors for HPAIV genesis” is referenced in line 208. The review “Pathobiological Origins and Evolutionary History of Highly Pathogenic Avian Influenza Viruses” by Dong-hun Lee is referenced in line 84.

  1. Changes in Hemagglutinin Receptor-Binding Specificity also leads to the genesis of HPAIVs. It needs to be discussed.

Changes in the receptor binding specificity of HA can indeed influence the virulence of AIVs in avian species. However, it has, to our knowledge, not been associated with the genesis of HPAIVs and thus would rather be classified as species-adaptation of AIVs to poultry. As there as many viral factors that influence the virulence of AIVs in avian hosts and the related species-adaptation and given that this aspect is outside of the scope of this review, we have resorted to shortly addressing them in the current manuscript (lines 202-218).

Reviewer 3 Report

Review of A.C.M de Bruin, et al., “Hemagglutinin Subtype-Specificity and Mechanisms of Highly

Pathogenic Avian Influenza Virus Genesis” (viruses-1759998).

In this manuscript, which is a detailed review and in silico analysis, Anja de Bruin, Mathilde Richard and colleagues explore a fundamentally interesting question in virology of highly pathogenic avian influenza virus (HPAIV) genesis –insertion(s) in the hemagglutinin (HA) gene leading to acquisition of a multibasic cleavage site (MBCS; also MCS, or polybasic cleavage site). This insertion occurs sporadically in nature in avian influenza virus (AIV) infection, particularly H5 and H7 subtypes of HA, and is a key virulence factor. A low-pathogenic AIV strain (LPAIV) that acquires an MBCS often becomes HPAIV in poultry, leading to severe HPAI disease and spillover among both wild and domestic birds. By analyzing AIV HA sequences, de Bruin et al. identified 51 apparently independent events resulting in acquisition of MBCS and generation of an HPAIV strain. They next examined the amino acid sequences of the MBCS insertions to map residues conferring increased protease cleavability; they found tetrabasic MBCS insertions with 4 lysine or arginine residues sometimes conferred HPAIV phenotype. Finally, they examined the hypotheses and RNA constraints for acquisition of MBCS by a combination of a thorough literature review, sequence analysis, and looking into RNA structures in the HA template during viral RdRP-mediated transcription. Their model presented (Fig. 5B) implying “trapping and stuttering” of the RdRP, resulting in addition of repeated RNA nucleotide sequences and the small number of insertions needed for a H5 or H7 to acquire MBCS (Fig. 4), is straightforward and ties the work together. They also examined alternative hypothesis for MBCS generation such as recombination for H7 strains, included in the Supplement. Overall, this is a deep and interesting exploration of one of the most important virulence factors in generation of HPAIV, a class of viruses that are pandemic and highly lethal in birds, and high risk for spillover into humans and other mammals.

A few points might be addressed to clarify the presentation, as follows:

1.     Table 1 and Fig. 3. China 1996-present Gs/Gd-H5 HPAIV and its descendants have spread across continents in both wild birds and poultry. As such this is an important lineage to examine, and the authors have discussed the literature. However they note that Gs/Gd-H5 MBCS are highly variable within this lineage and its descendants.

a)     It might be worth commenting on the degree of this variability, or adding a supplementary or in-text figure describing it. Are most of the Gs/Gd-H5 MBCS basically the same (from the P4 residue: RERRRKK-RG) and missing the P2 residue?

b)    Does the variation in Gs/Gd-H5 MBCS insertion sites support the trapping/stuttering model of MBCS acquisition (Figs. 5A and 5B)? Or an alternative model?

2.     The MBCS residue alignments presented in Fig. 3 and the Supplement are nicely presented and potentially useful for future researchers who study this question and as new strains containing MBCS are isolated. An open-access editable or downloadable file of these alignments (from Clustal?) in the Supplement would be a useful addition, rather than just an image in the text.

Author Response

We thank the reviewers for their constructive critiques. Please find their comments reproduced below, followed up by each of our answers in italics.

Comments and Suggestions for Authors: Reviewer 3

Review of A.C.M de Bruin, et al., “Hemagglutinin Subtype-Specificity and Mechanisms of Highly

Pathogenic Avian Influenza Virus Genesis” (viruses-1759998).

In this manuscript, which is a detailed review and in silico analysis, Anja de Bruin, Mathilde Richard and colleagues explore a fundamentally interesting question in virology of highly pathogenic avian influenza virus (HPAIV) genesis –insertion(s) in the hemagglutinin (HA) gene leading to acquisition of a multibasic cleavage site (MBCS; also MCS, or polybasic cleavage site). This insertion occurs sporadically in nature in avian influenza virus (AIV) infection, particularly H5 and H7 subtypes of HA, and is a key virulence factor. A low-pathogenic AIV strain (LPAIV) that acquires an MBCS often becomes HPAIV in poultry, leading to severe HPAI disease and spillover among both wild and domestic birds. By analyzing AIV HA sequences, de Bruin et al. identified 51 apparently independent events resulting in acquisition of MBCS and generation of an HPAIV strain. They next examined the amino acid sequences of the MBCS insertions to map residues conferring increased protease cleavability; they found tetrabasic MBCS insertions with 4 lysine or arginine residues sometimes conferred HPAIV phenotype. Finally, they examined the hypotheses and RNA constraints for acquisition of MBCS by a combination of a thorough literature review, sequence analysis, and looking into RNA structures in the HA template during viral RdRP-mediated transcription. Their model presented (Fig. 5B) implying “trapping and stuttering” of the RdRP, resulting in addition of repeated RNA nucleotide sequences and the small number of insertions needed for a H5 or H7 to acquire MBCS (Fig. 4), is straightforward and ties the work together. They also examined alternative hypothesis for MBCS generation such as recombination for H7 strains, included in the Supplement. Overall, this is a deep and interesting exploration of one of the most important virulence factors in generation of HPAIV, a class of viruses that are pandemic and highly lethal in birds, and high risk for spillover into humans and other mammals.

We thank the reviewer the kind words and positive assessment.

A few points might be addressed to clarify the presentation, as follows:

  1. Table 1 and Fig. 3. China 1996-present Gs/Gd-H5 HPAIV and its descendants have spread across continents in both wild birds and poultry. As such this is an important lineage to examine, and the authors have discussed the literature. However they note that Gs/Gd-H5 MBCS are highly variable within this lineage and its descendants.
  2. a)    It might be worth commenting on the degree of this variability, or adding a supplementary or in-text figure describing it. Are most of the Gs/Gd-H5 MBCS basically the same (from the P4 residue: RERRRKK-RG) and missing the P2 residue?

We thank the reviewer for this comment. It is indeed evident that there is a high degree of variation in the Gs/Gd-H5 lineage of HPAIVs, which probably stems from the extensive circulation of these HPAIVs in bird species. The original A/Goose/Guangdong/1996-virus contained the cleavage site motif PQRERRRKKR, but subsequent viruses contained alternative MBCS sequences due to deletions, substitutions, and sporadic insertions. The variation in the MBCS region, mainly representing that of Gs/Gd-lineage HPAIVs, has been reviewed and discussed previously in Luczo M et al. Rev. Med. Virol. 2015 (PMID:26467906) and we have changed the manuscript line 274 to refer to this review more explicitly, so that the readers of Viruses can quickly find that analysis. Furthermore, we have added a potential explanation of the high diversity in Gs/Gd-H5 MBCS motifs (lines 274-278). To answer the second question, based on the Luczo et al. review, Gs/Gd-H5 MBCS motifs mostly contain varying constellations of R and K residues in P5-P1, whereas non-basic amino acids are mostly located of downstream of P5. The P2 T residue from the LPAIV RETR consensus is indeed not present in the Gs/Gd-H5 HPAIVs.

  1. b)    Does the variation in Gs/Gd-H5 MBCS insertion sites support the trapping/stuttering model of MBCS acquisition (Figs. 5A and 5B)? Or an alternative model?

The MBCS motif of the A/Goose/Guangdong/1996 HA seems to have been generated through duplications (of the R AGA codon) or stuttering (presence of repeated stretches of As), which would be supported by our model. The MBCS sequence changes following this original conversion event mostly consist of nucleotide substitutions, but insertions and deletions have also been observed occasionally. Although we cannot provide at this stage experimental support for the application of our model to the continuous evolution of Gs/Gd-H5  HPAIVs MBCSs, it is in principle not incompatible, especially as the putative RNA stem-loop structure formed at the cleavage site that might lead to RdRp backtracking/stuttering has stabilized over time (Gultyaev et al. Virus Evol. 2019 PMID:31456885).

  1. The MBCS residue alignments presented in Fig. 3 and the Supplement are nicely presented and potentially useful for future researchers who study this question and as new strains containing MBCS are isolated. An open-access editable or downloadable file of these alignments (from Clustal?) in the Supplement would be a useful addition, rather than just an image in the text.

We agree with reviewer 3 and made the alignment file available as supplementary information.